# A Multi-Entity Knowledge Joint Extraction Method of Communication Equipment Faults for Industrial IoT

Kun Liang [1] , Baoxian Zhou [1],*, Yiying Zhang [1], Yeshen He [2], Xiaoyan Guo [3] and Bo Zhang [4]

1    College of Artificial Intelligence, Tianjin University of Science & Technology, Tianjin 300457, China;
     liangkun@tust.edu.cn (K.L.); yiyingzhang@tust.edu.cn (Y.Z.)
2    China Gridcom Co., Ltd., Shenzhen 518109, China; heyeshen@sgitg.sgcc.com.cn
3    Information and Communication Company, State Grid Tianjin Electric Power Company,
     Tianjin 300140, China; xiaoyan.guo@tj.sgcc.com.cn
4    State Grid Smart Grid Research Institute Co., Ltd., Nanjing 210003, China; zhangbo@geiri.sgcc.com.cn
*    Correspondence: zhoubaoxian@yeah.net

**Abstract:** The Industrial Internet of Things (IIoT) deploys massive communication devices for information collection and process control. Once it reaches failure, it will seriously affect the operation of the industrial system. This paper proposes a new method for multi-entity knowledge joint extraction (MEKJE) of IIoT communication equipment faults. This method constructs a multi-task tightly coupled model of fault entity and relationship extraction. We use it to implement word embedding and bidirectional semantic capture to generate computable text vectors. At the same time, a multi-entity segmentation method is proposed, which uses noise filtering to distinguish the multi-fault relationship of single corpus. We constructed a dataset of communication failures in power IIoT and conducted experiments. The experimental results show that the method performs best in tests with the Faulty Text dataset and the CLUENER dataset. In particular, the model achieves an F1 value of 78.6% in the evaluation of relationship extraction for multiple entities, and a significant improvement of 5–8% in its accuracy and recall. It enables effective mapping and accurate extraction of fault knowledge data.

**Keywords:** knowledge graph; entity recognition; relationship extraction; joint learning; multi-entity segmentation

## 1. Introduction

The industrial internet of things (IIoT) deploys a large number of perception and communication devices [1], which realize the real-time state perception of physical space–information space interaction and the normal operation of the system. Once a fault occurs, it is easy to cause problems, such as energy resource scheduling, distribution and out of control transmission. It will seriously endanger the operation safety of the integrated energy system [2]. Therefore, it is of great significance to accurately extract equipment fault information, locate the entity relationship of fault equipment, construct the fault knowledge graph of IIoT communication equipment [3,4] and realize real-time fault analysis and efficient troubleshooting of IIoT communication equipment.

Named entity recognition and relationship extraction are two key technologies to realize knowledge extraction [5]. Named entity recognition techniques are divided into three main approaches in terms of implementation techniques: rule-based, probabilistic graph and deep learning [6]. Rule-based learning relies more on manual or domain dictionaries, which not only lack flexibility, but also has poor recognition efficiency. With the development of probabilistic graphs, the learning of entity sequence information is carried out through directed and undirected graphs [7]. It reduces manual involvement and provides some improvement in efficiency, but generalization capability is poor and the efficiency of entity recognition needs to be improved [8]. Deep learning techniques build multiple

models to learn grammatical features through neural networks, which convert text into vectors covering contextual semantics [9]. It effectively improves on these shortcomings and greatly enhances the recognition of entities and relationship extraction [10–12]. The mainstream algorithms for relationship extraction tasks consist of two main types: manual annotation and remote supervision [13]. The former requires more human involvement but is also not generalizable, while the latter combines methods, such as deep learning to improve the accuracy and efficiency of relationship extraction [14,15]. In the process of domain-oriented knowledge extraction, deep learning greatly improves the generalization ability of knowledge extraction algorithms, which facilitates the process of building knowledge graphs in various domains, such as power [16] and geography [17].

Although the knowledge extraction effect of the current entity relation extraction model has been improved [18], the pipeline extraction method leads to the lack of relevance between entity extraction and relationship extraction [19,20], resulting in the loss of information. When facing the text of the fault of the IIoT communication equipment, because of the complex subordinate relationship between the equipment, confusion and misjudgment of the relationship between multi-fault entities can easily occur. The main reasons are: (1) when the traditional word embedding model maps the text, it is easy to cause the loss of time series information in the text [21]. Therefore, the relationship prediction cannot be well carried out in the follow-up tasks. (2) Most of the existing knowledge extraction is pipelined operation [22], and most of them establish independent models for entity extraction and relationship prediction [11–23]. Therefore, when facing the terms and relationships that appear in the professional field, the relational model cannot be combined with entity features to predict. This can easily lead to confusion in the labeling of multi-entity relationships [24–26].

This paper designs a knowledge extraction method for IIoT communication equipment faults, which can achieve fast and accurate indexing and localization for fault entities with relationship categories. The main contributions of this paper are as follows.

- We propose the ALBERT fault pre-training model for embedding the fault data, which solves the loss problem of fault text information.
- We build a joint coding model for fault text. The ALBERT-BilSTM network structure in the model enhances the coupling between subsystems, which effectively improves the accuracy of knowledge extraction.
- We have designed a joint extraction method of relationships between multiple entities. It can be simultaneously extracted on the relationship between multiple entities in a corpus.

The rest of the paper is organized as follows. Section 2 focuses on reviewing relevant research. Section 3 shows the system architecture and application scenarios of the joint extraction model, and the knowledge combination extraction algorithm is expounded, which is the main contribution of this article. Section 4 is the situational analysis of experimental simulations. Section 5 is a summary and a prospect for the next steps.

## 2. Related Work

Knowledge extraction is the basis for the construction of knowledge maps and is also the basis for knowledge visualization and recommended algorithms. Knowledge extraction usually contains two sub-tasks: named entity recognition and relationship extraction. In the above summary, we summarized the advantages and disadvantages and main branches of existing knowledge extraction techniques. Hereinafter, we focus on the research status of naming entity identification and relational extraction and discuss the differences between the algorithms proposed herein and existing algorithms [11,27–29].

In the aspect of named entity recognition, rule-based and statistics-based methods are mainly used in the early stage, but the portability is poor, and a lot of manpower is spent on tagging. With the development of machine learning, many existing models have significantly improved the efficiency of entity recognition, such as perceptron model, conditional random field model and so on. In recent years, deep learning-based methods

have become mainstream due to their excellent performance, for example, the bidirectional LSTM (long-and short-term memory) CRF model proposed by R. Grishman et al. [6]. The BiLSTM (bidirectional long short-term memory) model proposed by G. Lample et al. [7] integrates the self-attention mechanism, and CNNs (convolutional neural networks) process the word vector to identify military entities. The BiLSTM and CRF methods proposed by J. Li et al. [8] are used to extract entities from clinical symptoms of traditional Chinese medicine. The BiLSTM-CRF model is used to mark the motor entities sequentially [9]. In other fields, Kcm B et al. [17] have proposed a novel LSTM framework for short-term fog forecasting. Y. An. et al. [18] proposed a multi-head self-attention based bi-directional long short-term memory conditional random field (MUSA-BiLSTM-CRF) model, which is expected to greatly improve the performance of Chinese clinical named entity recognition.

In the aspect of entity relationship extraction, supervised relationship extraction is often used in the early stage. It includes the method based on CRF, but it needs a lot of manual labeling [11]. Y. Gu et al. [12] proposed a multi-instance, multi-label method to model relationship extraction and extract the situation, in which there may be multiple relationships in the entity pair. M. Surdeanu et al. [13] proposed the use of a multi-instance, multi-label and Bayesian network for relationship extraction. Although the effect has been improved, the number of model parameters has also greatly increased the training scale. On the other hand, the lightweight ALBERT [14] model proposed by Google can effectively reduce the number of model parameters, and the effect of knowledge extraction is better. In applications for power and beyond, S. Wang [15] and W. Shi [16] present new research ideas in terms of processing frameworks and hierarchical structures. B. Aha et al. [19] created a biomedical Knowledge Graph using Natural Language Processing models for relationship extraction. The generated biomedical knowledge graphs (KGs) are then used for question answering.

The proposed model designed in this paper uses a pre-trained model to reduce the number of training parameters and improve the representativeness of text information compared with the traditional model. Meanwhile, parameter sharing and multi-entity segmentation are used to achieve joint acquisition of fault knowledge. It provides data support and effective knowledge extraction means for forming fault database or fault knowledge graph [24]. It helps to assist fault repairers in decision making and rapid fault localization to improve the efficiency of fault investigation and diagnosis.

## 3. Multi-Entity Knowledge Joint Extraction Method

### 3.1. Architecture of Joint Fault Knowledge Extraction for IIoT Communication Devices

As shown in Figure 1, the industrial internet of things uses smart communication equipment to monitor and perceive the smart grid. Similarly, these communication devices constitute a power communication network with the power grid as the background. The power communication network can realize the fault detection and troubleshooting of the smart grid, which is also an indispensable part of the construction of the industrial internet of things.

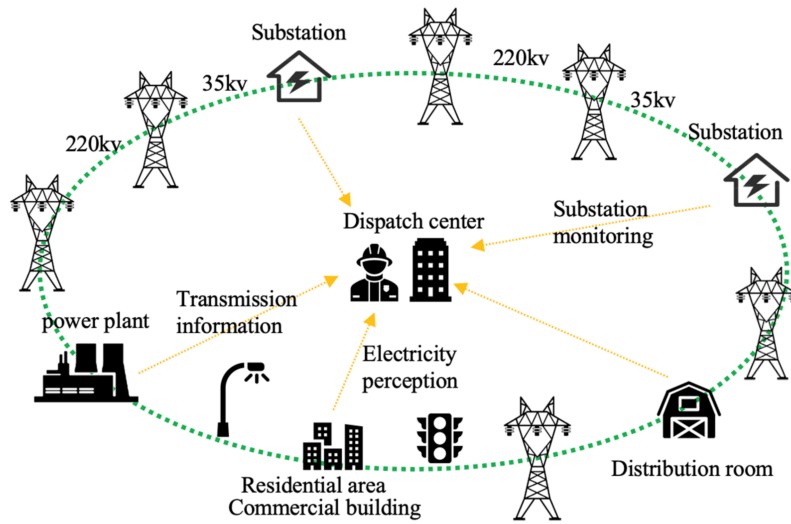

**Figure 1.** IIoT power communication equipment application scenario diagram.

The joint knowledge extraction model mainly includes three parts: data analysis, data processing and knowledge extraction. Among them, the data analysis part is to pre-process the fault information of IIoT communication devices into a corpus of short texts. The data processing part is to annotate the faulty entities and the entity relationships in the corpus to obtain the training set data. The word vector is generated by ALBERT and BiLSTM as word embedding models. Finally, CRF and entity segmentation are used in the knowledge extraction part to realize the joint extraction of faulty entities and entity relations. The main parts of the model are shown in Figure 2.

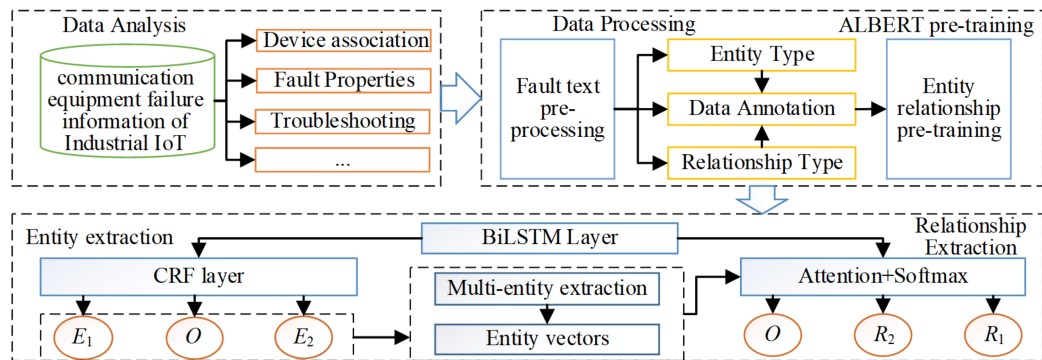

**Figure 2.** IIoT communication equipment fault knowledge extraction model diagram.

### 3.2. Fault Knowledge Text Pre-Training

Due to the various fault information of IIoT communication equipment, it is classified first. After that, the segmentation fault corpus is obtained by processing the sentence segmentation, removing the pause words and clauses. Finally, it is stored in the form of short text to realize the word embedding of the text. In this paper, the ALBERT pre-training model is used to represent the fault information by word embedding to realize the deep mining of fault features. The ALBERT model is used as a lightweight language pre-training model. As shown in Figure 3, it illustrates the pre-training structure of its word embedding representation in fault text processing.

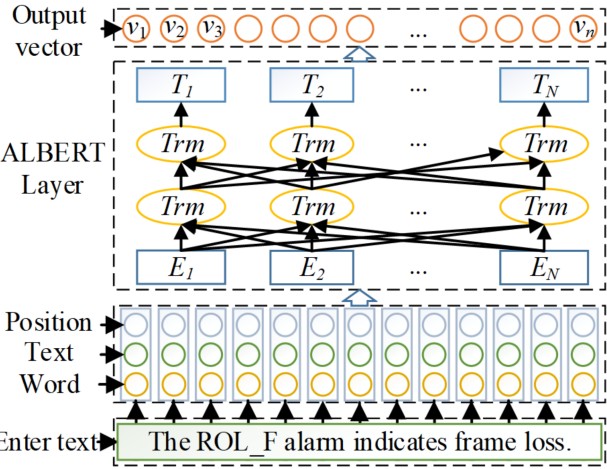

**Figure 3.** Faulty text pre-training structure diagram.

According to the input fault text content, there are three input vectors corresponding to the word vector, text vector and position vector. The output of the model is the word vector corresponding to each input word and contains the semantic information of the full text. The structure of the Transformer is shown in Figure 4, where Inputs represent the text vectors to be input. Input Embedding and Positional Encoding encode the input faulty text as a feed to the Transformer. The output of the multi-attention mechanism will be passed through ADD & Norm layers (ADD layer has the effect of concatenating the residuals of the two inputs and Norm stands for normalization). The output of this layer is passed to Feed Forward. Finally, the output of the ADD & Norm layer is again connected to other Transformer encoders or directly output as a word vector represented by each word of faulty text.

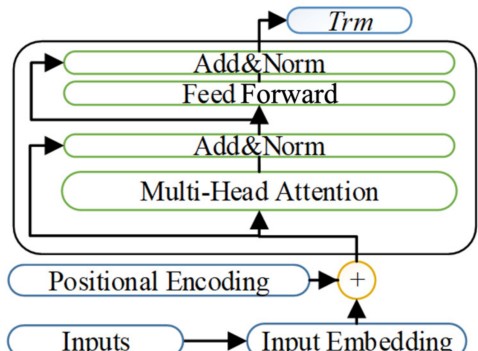

**Figure 4.** Transformer structure diagram.

*3.3. Parameter Sharing Code for Fault Knowledge Information*

Next, in-depth association mining is performed on entities and relationships in the context of the fault data text. This paper proposes a BiLSTM neural network coding structure with shared underlying parameters. It enables both tasks to update the shared parameters through the backpropagation algorithm. The role of this model is to strengthen the connection between subtasks and solve the problem of error propagation caused by the pipeline model. The information processing flow of the neurons in the coding layer in the BiLSTM neural network is shown in Figure 5.

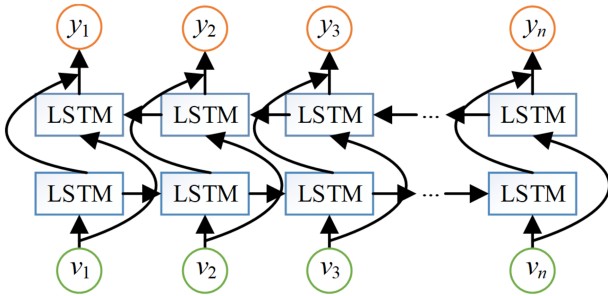

**Figure 5.** BiLSTM neural network structure diagram.

As shown in Figure 5, $v_n$ of the input LSTM neuron is the word vector matrix of the fault text. The output is the fault text feature vector $y_t$ after deep mining. The calculation process of the input gate is shown in (1), and the calculation of the output door $o_t$ and the forgetting gate $f_t$ is as shown in (2) and (3):

$$i_t = \sigma\left(W_i \cdot \left[y_{(t-1)}, v_t\right] + b_f\right) \tag{1}$$

$$o_t = \sigma\left(W_i \cdot \left[y_{(t-1)}, x_t\right] + b_o\right) \tag{2}$$

$$f_t = \sigma\left(W_f \cdot \left[y_{(t-1)}, v_t\right] + b_f\right) \tag{3}$$

At the same time, the forgetting door layer determines whether to transfer the input information $v$. The calculation process is shown in (4) and (5), where $W$ is the weight and $b$ is offset, and the final output hidden layer $y_t$ value and its calculation formula are shown in (6):

$$\widetilde{C}_t = \tanh\left(W_c \cdot \left[y_{(t-1)}, v_t\right] + b_c\right) \tag{4}$$

$$C_t = f_t \odot C_{(t-1)} + i_t \odot \widetilde{C}_t) \tag{5}$$

$$y_t = o_t \odot \tanh(C_t) \tag{6}$$

Through deep context feature extraction and parameter sharing coding, the weights and bias parameters of the two subtasks of entity labeling and relationship extraction are shared. This is conducive to the next step of entity sequence labeling and entity relationship extraction.

### 3.4. Multi-Entity Knowledge Joint Extraction

Through the pre-training of fault text corpus and the deep feature extraction of the BiL-STM neural network, a joint knowledge extraction model is constructed in this paper. The model is mainly divided into two parts: fault entity identification and entity relationship extraction.

Named entity recognition is used to mark entities through joint extraction of the CRF layer in the model. Entity relationship recognition uses multi-entity relationship recognition and the attention mechanism to realize relationship prediction. The frame diagram of the knowledge joint extraction model is shown in Figure 6.

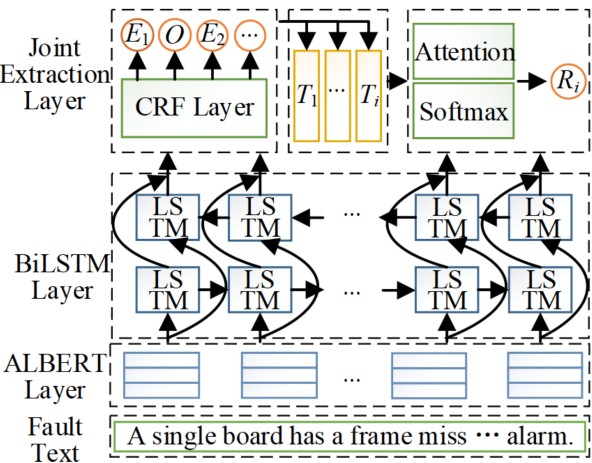

**Figure 6.** Multi-entity knowledge joint extraction framework diagram.

### 3.4.1. Fault Entity Identification

In order to realize the label prediction of fault entities, the CRF model is used to classify entity types in entity extraction. The text feature vector matrix $y_n$ obtained after ALBERT pre-training model and BiLSTM feature extraction is used as the input of the model to label the entity tag. Its structure diagram is shown in Figure 7.

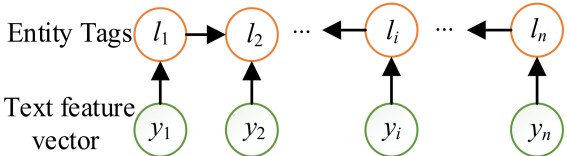

**Figure 7.** Entity marking algorithm diagram based on CRF.

We treat the fault data as an observation sequence $Y$ ($y_1, y_2, y_3, \ldots, y_n$), and each word in the observation sequence corresponds to a fault entity label $L$ ($l_1, l_2, l_3, \ldots, l_n$). $P(L \mid Y)$ is the conditional probability distribution of the label sequence $L$ when the observation sequence $Y$ is known.

Therefore, given the input observation sequence $Y$, we find the maximum possible entity label sequence $L$ by scoring the sentence sequences with the CRF model, as shown in Equation (7).

$$score(y, l) = \sum_{i=1}^{n} Y_{i,l_i} + \sum_{i=1}^{n+1} Q_{l_{i-1}, l_i} \tag{7}$$

where $Y_{i, l_i}$ represents the score of classifying word $y_i$ to the $l_i$ label. And $Q_{l_{i-1}, l_i}$ represents the transfer probability from the $l_i$ to $l_{i-1}$ node. When we predict the label of a word, we can use the previous labels that have been learned by the model. From Equation (7), it can be seen that the scoring value of the whole sequence is the sum of the scoring values of each position. Then, the Softmax layer is used for normalized classification, as shown in Equation (8):

$$P(l|y) = \frac{\exp(score(y, l))}{\sum_{l'} \exp(score(y, l'))} \tag{8}$$

Thus, the set with the maximum probability is selected as the final annotation sequence of the CRF layer. The output $y$ represents the fault entity label corresponding to the final input word vector.

### 3.4.2. Multi-Entity Relationship Extraction

This paper proposes a relationship extraction algorithm based on multi-entity segmentation. The algorithm realizes the division of multiple entity relationships in the same

sentence and solves the problem of multiple entity boundary confusion. The specific implementation process is shown in Figure 8.

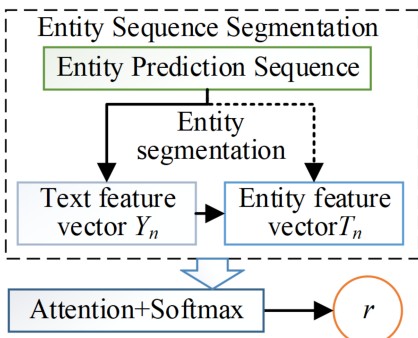

**Figure 8.** Algorithm diagram of multi-entity segmentation relation extraction.

As shown in Figure 9, the algorithm mainly includes two parts: segmentation input and relationship extraction. First, make statistics on the input entities. Then the entity segmentation algorithm is used for corresponding segmentation and extraction, and the feature vector $e_n$ of the corresponding entity is obtained. At the same time, the text feature vector $y_n$ and its co-coding are used as the input of the relationship extraction layer. Finally, in the relationship extraction layer, the Attention mechanism is used to re-measure the weight of the relationship between different entities and connect the Softmax layer to classify and predict the fault relationship. The specific algorithms of each layer are described below.

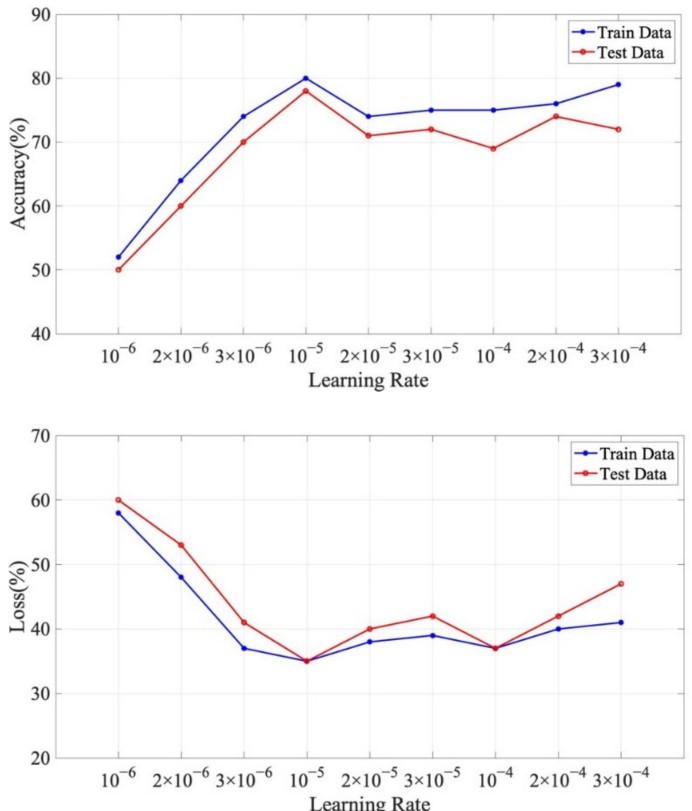

**Figure 9.** Average accuracy and loss of joint extraction model learning parameters.

Step 1: The entity segmentation algorithm is used to extract the entity feature vector pair $T_n$. The number n of fault entities in the current fault text is counted according to the

entity prediction label, and each fault entity feature vector is recorded as $e_i$. According to the combination number calculation formula, a total of $C_n^2$ pairs of corresponding entity feature vectors are extracted and recorded as $T_i$. It is calculated by Equation (9).

$$T_i = \left(e_i, e_j\right) \quad (\forall i, j \in n \ and \ i \neq j) \tag{9}$$

Step 2: The entity feature pair vectors $T_i$ and $Y_n$ are combined respectively, and the number of fault text feature vectors is $C_n^2$ as the input of the relationship extraction layer. Because the input length of the fault text is limited, the calculation loss is small, and it is convenient to adjust the weight of entity features in relationship extraction.

Step 3: The input fault text feature vector $T_i$ and $Y_n$ are used as the input of the Attention layer. The Attention layer is used to re-weigh the input feature weight and weight and output with each time series. Avoid the loss and dilution of semantic vector sequence information calculated by BiLSTM layer before. Emphasis is placed on the weight and bias of the relationship between fault entities. The calculation and adjustment is as follows:

$$C_i = \sum_{j=1}^{T} a_{ij} \cdot h_j \tag{10}$$

$$a_{ij} = \frac{\exp\left(e_{ij}\right)}{\sum_{k-1}^{T} \exp\left(e_{ij}\right)} \tag{11}$$

$$e_{ij} = \alpha\left(s_{i-1}, h_j\right) \tag{12}$$

where $c_i$ is the semantic coding vector, $a$ is the result of normalization of attention correlation coefficient $e_{ij}$, $h_j$ is the hidden layer output, $e_{ij}$ is the attention correlation coefficient, $\alpha$ is a calculation function and $s_{i-1}$ is the decoder output.

Step 4: The weighted sum vector obtained by the Attention mechanism is used as the text feature vector of the final relational prediction. Then, the Softmax function is accessed to predict the relationship. Finally, the entity relationship r in the case of maximum probability between any entities in the input text is obtained.

## 4. Experiment Analysis

### 4.1. Experimental Environment and Data

There are two sets of data sources for this paper. One set is the fault log data from the operation of the Southern China Power Grid, and also includes the power operation and maintenance manuals of the equipment and expert cases. It is taken from the alarm and performance event manual of the OptixOSN7500, a grid communication device. Through the preliminary manual screening, 15,000 sentences can be used in the experimental data. It includes 530 entities and 16 failure relationships. The other set is the Chinese dataset CLUENER, where the comparison experiments are conducted, which includes category labels for 10 life scenes and has a total corpus sample size of 12,091 items.

The configuration used in this article in the experimental environment is shown in the following Table 1.

**Table 1.** Experimental Environment.

| Project | Environment |
| --- | --- |
| Development language | Python3.7.5 |
| CPU | Intel i5-10210U CPU@2.4 GHz |
| GPU | GEFORCE RTX 2080 |
| Development framework | Tensorflow2.0 |
| Running memory | 16 GB |
| Hard disk size | 1 TB |

### 4.2. Data Preprocessing

First, the corpus is segmented, and missing words and blank lines are deleted. After that, the corpus is segmented and filled to a uniform word length. The word length is set to 200. Finally, the fault text data characteristics of IIoT communication equipment are combined. Some entities and relationships are shown in Tables 2 and 3.

**Table 2.** Fault Entity Corpus Tagging.

| Identification | Category | Example |
|:---:|:---:|:---:|
| N | Equipment name | Ethernet veneer, optical fiber, pluggable optical module |
| C | Issue cause | Aging, shedding, pressure loss |
| P | Failure phenomenon | Alarm, interruption, packet loss |
| M | Processing method | Check, connect, clean |

**Table 3.** Entity Relation Corpus Tagging.

| Identification | Category | Example |
|:---|:---:|:---:|
| U | Include | Ethernet veneer with interface board. |
| S | Causality | Cause. |
| T | Constellation | And. |

This article uses the "BIO" label strategy. Mark each element as "B-X", "I-X" or "O". Among them, "B" represents the beginning of the marked entity or relationship, "I" represents the middle, and "O" indicates that the text does not belong to any type of entity or relationship. Table 4 shows an example of fault corpus tagging.

**Table 4.** Training Set Corpus Sample Labeling Example Table.

| No. | Character | Label | No. | Character | Label |
|:---|:---:|:---:|:---|:---:|:---:|
| 1 | the | O | 9 | the | O |
| 2 | frame | B-P | 10 | veneer | B-N |
| 3 | miss | I-P | 11 | can | O |
| 4 | alarm | I-P | 12 | easily | O |
| 5 | in | O | 13 | lead | B-S |
| 6 | data | O | 14 | to | I-S |
| 7 | transmission | O | 15 | service | B-P |
| 8 | from | B-S | 16 | interruption | I-P |

### 4.3. Evaluation Index

In this paper, the accuracy P (Precision), recall rate R (Recall) and F1 value (F1-score) are used as the evaluation criteria for the extraction results of entities and relationships.

$$P = \frac{TP}{TP + FP} \tag{13}$$

$$R = \frac{TP}{TP + FN} \tag{14}$$

$$F1 = \frac{2PR}{P + R} \tag{15}$$

In the above formula, TP (True Positive) indicates the number of positive samples predicted as positive samples. TN (True Negative) means to predict the number of negative samples as negative samples. FP (False Negative) means to predict the number of positive samples as negative samples. Finally, calculate the F1 value according to Equation (15). The higher the value, the better the classification effect.

### 4.4. Model Parameter Training

In terms of experimental parameters, this article chose different parameter settings for comparison. Finally, we determine the best parameter settings, where the dimension of the word vector is 200, the size is 256, the number of layers of the BiLSTM model is 2, the value of dropout is 0.8, the head h of self-attention is set to 6, and the bias variable is set to 10.

In the experimental part, we optimize the joint extraction model by adjusting the learning rate and the number of iterations. Among them, the learning rate will directly affect the speed and accuracy of the model convergence. When the learning rate is set too large, although the model learning speed is fast, it may cause oscillations (A too large learning rate setting leads to a non-convergence of the training process, and parameters such as accuracy show large fluctuations during the training process.). If the learning rate is too small, it may fall into a local optimum (A too small learning rate setting will result in a recent extreme point in the training process and cannot jump out of the local range to achieve the overall optimal.). Similarly, the number of iterations represents the entire process of model training. When the number of iterations is too few, the model training will not be mature enough, which will lead to a larger loss value, and too many iterations will cause over-fitting and reduce the accuracy rate. Among them, Acc_train represents the accuracy of the training set, Loss represents the verification error and Acc_test represents the accuracy of the test set.

As shown in Figure 9, when the learning rate is $1 \times 10^{-5}$ and $2 \times 10^{-4}$, the accuracy of the model training set is higher, but the prediction effect of the latter verification set is slightly lower. Similarly, the loss rate of the model is lower when the learning rate is $1 \times 10^{-5}$ and $1 \times 10^{-4}$. In the case of taking into account both the accuracy of model knowledge extraction and the loss rate, $1 \times 10^{-5}$ is the best learning rate.

As shown in Figure 10, according to the trend of the line chart, the accuracy of the model improves rapidly in the training processes, and the accuracy is the highest when the number of iterations reaches 70, and then tends to be stable. The model loss rate varies with the number of model training iterations. As can be seen from the diagram, the loss rate of the model decreases rapidly in the iterations. When the number of training is 70, the loss effect of training set and verification set is better. When the training time is 80, the loss rate of the training set is the lowest, but the loss of the verification set is larger. The iterative effect of comprehensive accuracy and loss rate preserves the model parameters with training times of 70.

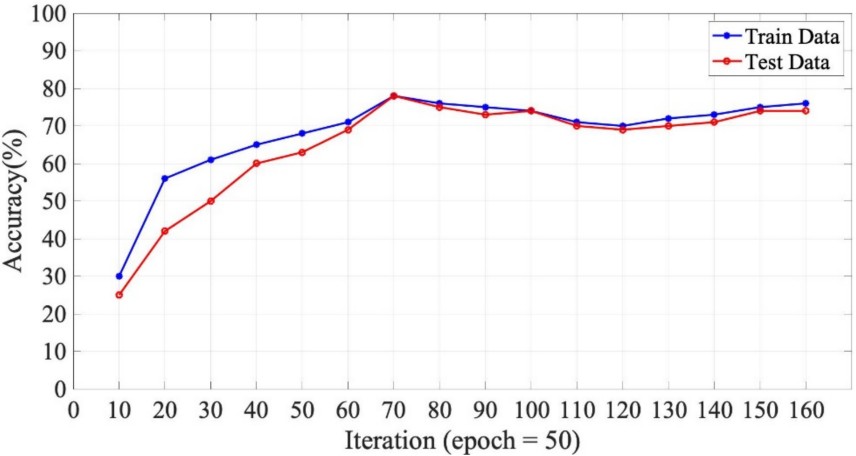

**Figure 10.** *Cont.*

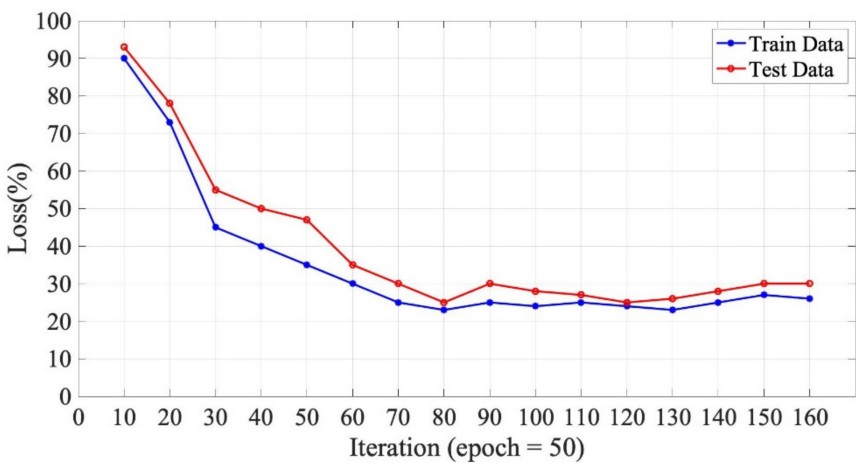

**Figure 10.** Average iterative accuracy and loss of model training.

*4.5. Comparative Analysis of Model Results*

In order to verify the model experimental effect, four groups of comparison experimental groups are set up in this paper: BiLSTM-CRF [11], LSTM-Attention-CRF [27], BERT-LSTM-CRF [28] and BERT joint extraction model [29] as knowledge extraction effect control. The experiments have compared the CLUENER dataset and the Failure Text dataset of industrial network communication equipment.

Table 5 shows the evaluation results for named entity recognition of device fault text with the CLUENER dataset by various classical models. As we can see from the table, the left side shows the evaluation results of named entity recognition for faulty text, and the performance of the MEKJE model proposed in this paper is on average 3% better in terms of both accuracy, recall and F1 value compared with models such as BiLSTM-CRF. Our comparison experiments with the CLUENER dataset on the right side show that the MEKJE model also performs well, with an average improvement of 5% in F1 values compared to other models. This also proves that the pre-trained model can produce a greater help in the named entity recognition accuracy improvement. Although the model named entity recognition results are similar to the Bert model, the advantage of this model is the reduction of training parameters and the significant reduction of training cost. This also shows that the model is not only for the electric power field, but also for other datasets with good recognition results and strong generalizability.

**Table 5.** Fault Entity Label Prediction.

| No. | Model | Fault Text Dataset | | | CLUENER Dataset | | |
|---|---|---|---|---|---|---|---|
| | | **P** | **R** | **F1** | **P** | **R** | **F1** |
| 1 | BiLSTM-CRF | 0.658 | 0.613 | 0.634 | 0.685 | 0.644 | 0.673 |
| 2 | LSTM-Attention-CRF | 0.731 | 0.663 | 0.695 | 0.709 | 0.724 | 0.711 |
| 3 | BERT-LSTM-CRF | 0.759 | 0.697 | 0.726 | 0.743 | 0.716 | 0.721 |
| 4 | BERT joint extraction | 0.779 | 0.765 | 0.771 | 0.858 | 0.842 | 0.860 |
| 5 | MEKJE | 0.773 | 0.756 | 0.764 | 0.943 | 0.916 | 0.943 |

As shown in Table 6. The MEKJE model proposed in this paper is higher than other classic models in relational extraction. Whether in the Fault Text dataset or the CLUENER dataset, the model's performance is the best. In particular, it has increased 6.1% and 6.2% in the precision and recall compared with the optimal BERT joint extraction model, and the F1 value reached 78.6%. In order to verify the generality of the model, the experiment also tests on the CLUENER dataset. The experimental results show that the MEKJE model also achieves the best results on the open-source Chinese dataset. It increased by 8.3%, 6.9% and 8.3% on the accuracy, recall and F1 score. Experiments show that the MEKJE model

can deeply explore the relationship between the multi-entity entities, improve the accuracy of the relationship extraction.

**Table 6.** Multi-entity Relationship Extraction Prediction.

| No. | Model Name | Fault Text Dataset | | | CLUENER Dataset | | |
| --- | --- | --- | --- | --- | --- | --- | --- |
| | | P | R | F1 | P | R | F1 |
| 1 | BiLSTM-CRF | 0.549 | 0.498 | 0.522 | 0.571 | 0.537 | 0.553 |
| 2 | LSTM-Attention-CRF | 0.695 | 0.603 | 0.645 | 0.662 | 0.548 | 0.595 |
| 3 | BERT-LSTM-CRF | 0.706 | 0.591 | 0.643 | 0.744 | 0.714 | 0.728 |
| 4 | BERT joint extraction | 0.735 | 0.713 | 0.723 | 0.860 | 0.825 | 0.842 |
| 5 | MEKJE | 0.796 | 0.778 | 0.786 | 0.894 | 0.860 | 0.876 |

Figure 11 shows the performance of each model in terms of entity and relationship extraction accuracy, where the solid line represents the entity and relationship extraction accuracy for the Faulty Text dataset. The dashed line represents the entity and relationship extraction accuracy for the CLUENER dataset. As can be seen from the graph, the entity extraction accuracy of the MEKJE model is almost the same as that of the BERT joint extraction model, while the accuracy of relational extraction is significantly higher than that of the latter.

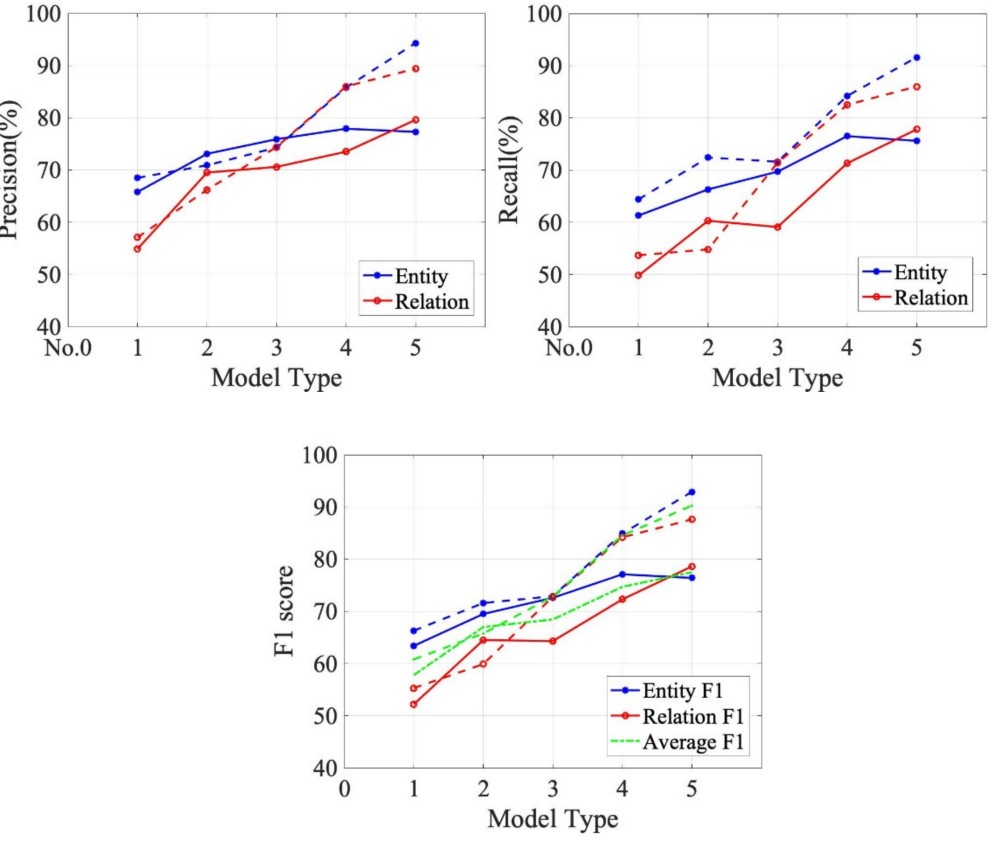

**Figure 11.** Evaluation of MEKJE model in comparison with classical models.

Similarly, it shows the performance of each model in terms of recall rates for entity and relationship extraction. As can be seen from the graph, the entity extraction recall rate of the MEKJE model is slightly lower than that of the BERT joint extraction model, while the accuracy of relational extraction is much higher than the latter.

By calculating the accuracy of the model and the recall rate, a comprehensive assessment of each model can be obtained for F1 values. As shown in Figure 11, the blue line

represents the F1 value extracted by the entity, the red line represents the F1 value extracted, and the green line represents the average F1 value of each model. In the comparison of the entity relationship with several classic models, this model is optimized by the entity relationship with several classic models, and improves model application efficiency.

### 4.6. Analysis of Sample Prediction Results

According to the prediction results in Table 7, the input sample will output the prediction results of the fault entity and relationship after knowledge extraction through the model. According to the predefined entity relationship label, the number and name of the faulty entity in the model output text. At the same time, the prediction of the relationship between the two entities is realized and expressed in the form of combination.

**Table 7.** Experimental Prediction Results of Joint Extraction Model.

| No. | Input Corpus | Prediction Result |
|---|---|---|
| 1 | There is an error alarm on the service transmission optical path, which may be caused by the inconsistent configuration of the time slots bound at both ends, or the veneer failure caused by the performance degradation of the service transmission optical path. | {'Equipment name':[N1:' service transmission optical path ', N2:' veneer ', N3:' time slots '], 'Issue cause':[C1:' Inconsistent configuration ', C2:' performance degradation ',], 'Failure phenomenon':[P1: error alarm ], 'Causality':[S1: (P1, C1),S2: (P1, C2)], 'Constellation': [T1: (C1, C2)]} |
| 2 | Loose fiber optic connectors, unclean fiber optic connectors, fiber optic cable failures, and poor contact of components in optical path transmission, etc., will all cause deterioration of transmission performance. | {'Equipment name':[N1:' fiber optic connectors ', N2:' optic cable '], 'Issue cause': [C1:' loose ', C2:' failures', C3:' poor contact '], 'Failure phenomenon':[P1:' deterioration of transmission performance '], 'Causality':[C1: (P1,C1), C2: (P1,C2), C3: (P1,C3)]} |
| 3 | For Ethernet boards with interface boards, replace the interface boards first. For Ethernet boards without interface boards, directly replace the alarm boards. | {'Equipment name':[N1:' Interface board ', N2:' Ethernet boards ',N3: ' veneer ' ,N4:' alarm boards '], 'Processing method': [M1:' replace ', M2:' replace '], 'Include':[U1: (N1, N2), U2: (N1, N3)]} |
| 4 | For the ring multiplex section, check from the east-west fiber connection. For the linear multiplex section, check the connection of the working protection optical fiber. | {'Equipment name':[N1:' ring multiplex section ', N2: ' east-west fiber ', N3: ' linear multiplex section ', N4: ' working protection optical fiber '], 'Processing method':[M1:' Check ']} |
| 5 | The multiplex section status indication alarm means that an event such as fiber disconnection or terminal node failure has occurred, which may trigger multiplex section protection switching. | {'Equipment name':[N1:' multiplex section ', N2:' terminal node '], 'Issue cause':[C1:' fiber disconnection ', C2:' failure '], 'Failure phenomenon':[P1:' status indication alarm ', P2: ' protection switching '], 'Causality':[C1: (P1,P2)]} |
| 6 | For the two-fiber ring multiplex section and linear multiplex section, if fiber disconnection and other failures occur again. | {'Equipment name':[N1:' two-fiber ring multiplex section ', N2:' linear multiplex section '], 'Issue cause':[C1:' fiber disconnection ', C2:' other failures '], 'Failure phenomenon':[P1: ' Business interruption '], 'Causality':[C1: (C1, P1), C2: (C2,P1)]} |

Figure 12 shows the fault knowledge graph constructed from the entities and relationships of the IIoT communication device faults. The knowledge extraction of the fault text is performed by means of a federated model, and we will simulate and model the resulting entities and relations by means of Neo4j. Entities are represented by dots and relationships are represented by arrow connecting lines between two entities. For example, pink represents the fault phenomenon, blue represents the fault treatment and orange represents the cause of the fault.

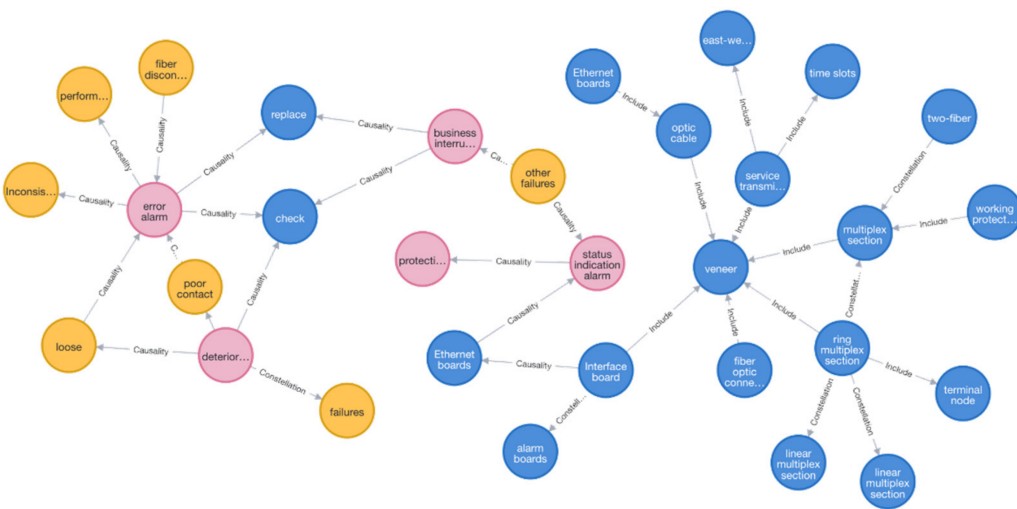

**Figure 12.** Fault knowledge graph of IIoT.

When an equipment fault occurs, the maintenance worker can refer to the power knowledge map to search for fault handling cases and the relationships between related equipment to locate the cause of the fault as quickly as possible. Thus, the knowledge map shows the correlation between equipment more vividly, which is helpful for fault handling.

## 5. Conclusions

The construction of fault knowledge graph of IIoT communication equipment has significant development potential and application value for realizing the stable operation of integrated energy system. In order to realize fault knowledge extraction, this paper proposes a joint fault knowledge extraction method based on the fault data of existing IIoT communication equipment. The model is based on the ALBERT pre-training model and BiLSTM neural network to embed text words to generate text vectors. Secondly, the entity prediction label and multi-entity segmentation algorithm based on CRF layer realize the joint extraction of fault entity and entity relationship. The experimental results show that the F1 value of this method reaches 78.6%, which increases the prediction accuracy by 2% and the recall rate by 3.8% compared with classical BERT-LSTM-CRF and other models. Through comparison experiments on the CLUENER dataset, it is directly demonstrated that the method can achieve more accurate label prediction of equipment fault entity relationships. It provides a data extraction method for establishing a more complete knowledge graph of industrial IIoT communication equipment faults, which can effectively improve the automated operation and maintenance of energy integrated systems.

**Author Contributions:** All authors contributed to the writing and revisions; writing—review and editing, K.L.; writing—original draft, B.Z. (Baoxian Zhou); conceptualization, Y.Z.; data curation, Y.H.; formal analysis, X.G.; resources, B.Z. (Bo Zhang). All authors have read and agreed to the published version of the manuscript.

**Funding:** This research was funded by the National Natural Science Foundation of China, grant number 61807024.

**Data Availability Statement:** Data available on request due to restrictions of privacy or ethical.

**Conflicts of Interest:** The authors declare no conflict of interest.

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
