# Peer review of "A Multi-Entity Knowledge Joint Extraction Method of Communication Equipment Faults for Industrial IoT"

_electronics, doi:10.3390/electronics11070979_

Round 1
Reviewer 1 Report
Authors propose to use a combination of transformer neural networks and recurrent neural networks for identification of fault messages in text streams.
While topic is actual, the manuscript have lots of issues which should be addressed first:
* Images should be re-done by saving those to eps or equivalent to be 300+dpi
* Formulas should be updated to common notation "*" - convolution, ⦿ - pointwise multiplication, and none for matrix multiplication.
* Formula (7), there is no x, which x -> infinity
* Formula (8) lower indexes
* Authors presenting accuracy statistics for binary classification, but using multi-class classification formulation, definitions of global F1 score, and multiclass confusion matrix should be added to the manuscript.
* No information how many epochs were used (maybe authors do not know the difference between the epoch and iteration) The iterations must be provided in the figures!
* No experimentation on not-fault data messages. How likely the false-positive classification would appear if random text without fault would be runned with a given model?
The current state of manuscript is not sufficient to make an assessment, which can only be done after the major revision.
Reviewer 2 Report
The paper proposes a novel approach to multi-entity knowledge joint extraction (MEKJE) of IIoT communication equipment faults. Through empirical evaluation, the authors demonstrated that the proposed model achieves fast and accurate indexing and localization for fault entities with relationship categories. While I find the work interesting, I also identify various issues as detailed below.
Section 2:
- This section is too brief, and I wonder if the authors still want to leave it as a single section. If it is the case, please consider revising the section to bring in some background related to the current work. For instance, it is necessary to provide information about LSTM, BiLSTM, etc.
Section 3:
- Please provide a short description for ALBERT.
Section 4:
- When reading the first paragraph, I think that more details about the dataset could be useful to help readers understand how it was collected/labeled.
- It seems the definition of false positive (FP) in Line 266 is not correct. According to the definition, actually this is false negative. Please consider correcting this.
- Please provide a short description for oscillations and local optimum (Line 278).
- In Section 4.5, the authors compared their approach with some baselines. It is not clear where these baselines come from. Have they been used for the same purpose? Moreover, there are no references to the original work.
- It is highly advisable to add more descriptions for the experimental results, especially those presented in Table 5, 6.
- It is not clear what the functions of the fault knowledge graph shown in Figure 14 are. What can an administrator learn from the graph? Please add more detail to the figure.
Related Work:
- One of the most critical points is that the paper does not have a Related Work section. Such a section is important since it associates the proposed approach with exiting studies, as well as provides background for the whole paper. At its current form, the paper does not explain explicitly the state-of-the-art research, and how the authors bridge the gap. In this section, I expect to see the application of LSTMs in other domains. Therefore, I would suggest the authors review more studies, and some of them are the following papers: (1) A survey on LSTM memristive neural network architectures and applications (https://link.springer.com/article/10.1140/epjst/e2019-900046-x); (2) Unavailable Transit Feed Specification: Making It Available With Recurrent Neural Networks (https://ieeexplore.ieee.org/abstract/document/9345512); (3) Application of LSTM for short term fog forecasting based on meteorological elements (https://www.sciencedirect.com/science/article/abs/pii/S0925231220304884).
MINOR REVISIONS:
- It is necessary to provide the full name of CRF by its fist occurrence. Currently, it is not clear what it really means.
- Line 103: "the Figure 1" --> "Figure 1"
- Line 161: "The it calculation" --> "The calculation"
- Line 271: "Finally determine the best parameter settings" --> "Finally, we determine the best parameter settings"
- Elsewhere: I would avoid using references in the following way: "[14] proposes to use multi-instance, ..." --> It is better to refer to an existing study as follows "Surdeanu et al. [14] propose using multi-instance, ..."
Round 2
Reviewer 1 Report
All equations should be careful reviewed!
Equations (5) - (6) - pointwise multiplication sign is missing, should be fixed
Eq (7) second sum no initial index i, Q have no indexes too,
Eq (8) have no sumation indexes, now it's constant - huge error
Eq (11) summation index wrong, should be fixed
Reviewer 2 Report
I thank the authors for their effort to address my comments. Compared to its previous version, the paper is now improved a lot, and therefore, I happily recommend the acceptance.
Author Response
Dear Reviewer,
Thank you very much for your recommendation.
Best regards,
Baoxian Zhou